# Current Treatment of Myasthenia Gravis

**DOI:** 10.3390/jcm11061597

**Published:** 2022-03-14

**Authors:** Mohammed K. Alhaidar, Sumayyah Abumurad, Betty Soliven, Kourosh Rezania

**Affiliations:** Department of Neurology, University of Chicago Medical Center, Chicago, IL 60637, USA; sumayyah.abumurad@uchospitals.edu (S.A.); bsoliven@neurology.bsd.uchicago.edu (B.S.); krezania@neurology.bsd.uchicago.edu (K.R.)

**Keywords:** myasthenia gravis, pyridostigmine, beta adrenergic, terbutaline, corticosteroids, prednisone, tacrolimus, azathioprine, cyclosporine, methotrexate, cyclophosphamide, stem cell transplant, eculizumab, efgartigimod, intravenous immunoglobulin, plasma exchange

## Abstract

Myasthenia gravis (MG) is the most extensively studied antibody-mediated disease in humans. Substantial progress has been made in the treatment of MG in the last century, resulting in a change of its natural course from a disease with poor prognosis with a high mortality rate in the early 20th century to a treatable condition with a large proportion of patients attaining very good disease control. This review summarizes the current treatment options for MG, including non-immunosuppressive and immunosuppressive treatments, as well as thymectomy and targeted immunomodulatory drugs.

## 1. Introduction

Myasthenia gravis (MG) is a an acquired, autoimmune disease of the neuromuscular junction which is caused by autoantibodies against different components of the neuromuscular junction [1]. Prior to the introduction of acetylcholinesterase inhibitors in 1934, patients diagnosed with MG had a grave prognosis and many succumbed to respiratory failure and pneumonia in 1–2 years [2,3]. The discovery of anticholinesterase (AChE) inhibitors resulted in improved diagnostic accuracy and also decreased mortality, which was estimated at 32% in 6 years in 1953 [4]. Thymectomy was introduced in 1939 and its role in MG pathogenesis was later demonstrated [5,6], but its effect in MG was not confirmed through a randomized clinical trial until decades later [7]. Corticosteroid treatment for MG was introduced in the 1960s [8,9] followed by the use of azathioprine, plasma exchange (PLEX) and intravenous immunoglobulin (IVIG) [10,11,12,13,14,15,16]. These treatments along with the availability of antibiotics and advanced respiratory care have led to substantial improvement in quality of life and a mortality rate of 5–9% [17]. Treatment of MG remains challenging as a subgroup of patients are treatment refractory, therefore having recurrent hospitalizations for MG crisis, requiring maintenance IVIG or PLEX. Therefore, more aggressive approaches such as “rebooting” of the immune system with high-dose cyclophosphamide or autologous bone marrow transplantation were used in some refractory cases with life-threatening disease [18,19]. On the other hand, as MG is usually a chronic disease, side effects of exposure to chronic use of steroids or other immunosuppressants may dramatically affect the lifespan or quality of life.

## 2. Non-Immunosuppressive Treatments

### 2.1. Acetylcholinesterase (AChE) Inhibitors

Peripherally acting AChE inhibitors are used as symptomatic treatments for temporarily alleviating muscle weakness in MG patients. They work by reversibly inhibiting the action of AChE, preventing the breakdown of acetylcholine (ACh) and, thus, increasing the amount of ACh available at the neuromuscular junction (NMJ) to bind to postsynaptic ACh receptors [20]. AChE inhibitors have remained the standard management for MG since early observations of dramatic response of these drugs (physostigmine, neostigmine) in the mid-1930s [21,22,23]. Clinical response to AChE inhibitors may vary between individuals or the muscles involved; for example, patients with ocular MG may have better alleviation of the ptosis than diplopia [24,25]. Oral pyridostigmine has been the most widely used preparation since the 1950s among other AChE inhibitors due to its longer duration of action, better tolerance profile, and fewer cholinergic side effects [26,27,28,29,30,31]. It is usually given at an initial dose of 30–60 mg every 4–6 h and may be increased to 90–120 mg every 4–6 h based on patient response and tolerance. It has an onset of action as early as 15–30 min with a duration of about 3–4 h [32]. A sustained-release form of pyridostigmine (Mestinon Timespan 180 mg) at nighttime may be useful for patient with weakness upon awakening. It is suggested to discontinue AChE inhibitors in patients with MG crisis requiring mechanical ventilation support, due to concerns of increased bronchial secretion and bronchospasm, with a goal of restarting them during the waning process or after extubation. An intravenous preparation of pyridostigmine (1 mg IV equivalent to 30 mg PO) may be considered in selected settings where intravenous immunoglobulin or plasma exchanges are unavailable, but caution is needed as there may be an increased risk of cardiac arrythmia [33,34]. Side effects are either due to stimulation of the ACh muscarinic receptors, which include gastrointestinal disturbances (abdominal cramps, diarrhea, nausea, increased salivation), increased bronchial secretions, lacrimation, hyperhidrosis and bradycardia, or stimulation of nicotinic receptors, including muscle cramps, and fasciculations. Muscarinic side effects may be mitigated with the use of loperamide or glycopyrrolate. AChE inhibitors may be discontinued once clinical remission is achieved or if patients develop incapacitating side effects [35]. Paradoxical weakness (such as myasthenia gravis exacerbation) may occur with very high doses of AChE inhibitors [36].

### 2.2. β-Adrenergic Agonists

Ephedrine, a sympathomimetic, was used as a treatment for MG in the 1930s, but it is rarely used in the present day [37,38]. β-adrenergic agonists increase cyclic AMP in the muscles and lymphocytes, which may lead to symptomatic relief of fatigue as well as regulatory effects on lymphocyte proliferation and antibody synthesis [39,40]. The severity of experimental autoimmune MG (EAMG) is increased by sympathectomy and alleviated by terbutaline, a β2 adrenergic agonist [41]. In a small randomized, double-blind placebo-controlled crossover pilot study, we found that 63% of patients had improvement of quantitative MG score of 3.0 or greater during the terbutaline phase [42]. Interestingly, salbutamol has been shown to enhance neuromuscular junction synaptic structure by counteracting the long-term effect of cholinesterase inhibitors on synaptic area in a transgenic model of AChR deficiency [43]. These findings have implications in both genetic and autoimmune myasthenia. Thus, β2 adrenergic agonists can be considered in patients who cannot tolerate cholinesterase inhibitors, in the presence of relative contraindications (such as severe asthma), or loss of response to cholinesterase inhibitors over time.

## 3. Immunosuppressive Treatments

### 3.1. Corticosteroids

Corticosteroids are the first-line immunosuppressant therapy for patients with MG who remain symptomatic while on AChE inhibitors or those who desire better symptom control. Early use of oral steroids in patients with pure ocular symptoms may delay or reduce the risk of generalization and worsening of the underlying symptoms [44,45,46,47,48]. Moreover, earlier initiation of steroids therapy during the disease course may allow for early and long-term remission, with 70–80% of patients on steroids achieving marked improvement or complete resolution of symptoms as opposed to 10–20% who achieve spontaneous remission [17,49,50,51]. Oral corticosteroids have a rapid therapeutic onset with clinical improvement that may be observed within two weeks after initiating therapy, with most improvement seen over the first 4–8 weeks [51]. Approximately 20–40% of MG patients who are started on steroids may experience varying degrees of transient worsening of weakness that may develop within the first two weeks of therapy initiation, particularly in patients started on higher steroids doses, in older individuals, and in thymoma-associated or early-onset MG [52,53]. Starting treatment with low-dose prednisone (≤25 mg) with gradual escalation has been suggested to avoid steroid-induced paradoxical weakness [54]. As such, a low dose and slow titration approach in patients with mild to moderate oropharyngeal or respiratory symptoms is recommended in outpatient settings to avoid the transient weakness worsening—initial prednisone dose of 5–10 mg daily with a gradual dose escalation of 5 mg every 7–10 days up to 60–80 mg daily or until desirable symptom control is reached. A faster dose escalation with close monitoring can be employed in patients who do not have significant respiratory or bulbar weakness and need higher doses to control the symptoms. Others have used alternate day dosing of prednisone, staring at 10 mg, increasing in 10 mg increments to 100 mg or to 1.5 mg per kilogram of body weight on alternate days [7]. Additionally, starting lower doses may be sufficient in patients with only ocular or mild symptoms. Once the desired symptom control is achieved, the dose should be tapered very slowly every month to the minimum effective dose. Once daily dose is <15 mg, a reduction of 1 mg per month may be attempted to avoid symptom relapse. In contrast, patients with severe myasthenic symptoms in inpatient settings or in the intensive care unit may benefit from starting higher prednisone doses for earlier symptom control while on bridging therapy. In a prospective study, 45 patients with grades IIa to V according to the Myasthenia Gravis Foundation of America (MGFA) were treated with intravenous immunoglobulin (IVIG) 2 g/kg divided over 5 days followed by therapeutic dose of steroids (1 mg/kg/day or 0.75 mg/kg/day in patients with comorbidities). Seven to 10 days after the IVIG course, only 2.2% of the patients had an MG exacerbation [55]. Caution is to be exercised in applying this approach to every MG patient as most subjects in that study had a milder phenotype, i.e., more than half of the subjects in class IIa and IIb, and ~15% in classes IV and V. Side effects, particularly with the long-term use of prednisone, include hypertension, obesity, gastric and peptic ulcers, cataracts, cushingoid appearance, psychological disturbances, opportunistic infections, sepsis, and serum electrolyte derangements [50]. It is recommended to test for tuberculosis (Quantiferon Gold test) before starting treatment, and administer prophylactic tuberculosis treatment if there is evidence of previous exposure [56]. The authors generally start patients on daily calcium and vitamin D supplementation as well as a proton pump inhibitor and recommend yearly bone density testing in patients on chronic treatment.

### 3.2. Azathioprine

Azathioprine is a pro-drug of 6-mercaptopurine, which is then metabolized to 6-thioguanine, which interferes with DNA synthesis by inhibiting the biosynthesis of purine nucleotides in rapidly proliferating cells such as T- and B-lymphocytes [57,58,59]. Other metabolic pathways of 6-mercaptopurine include degradation to 6-thiouric acid through xanthine oxidase, and methylation to 6-methyl mercaptopurine through thiopurine methyltransferase (TPMT) [59] (Figure 1). Azathioprine is one of the most widely used steroid-sparing drugs for MG. However, it has a delayed therapeutic onset with initial clinical improvement that may be experienced four to six months after therapy initiation but can take up to 18 months before experiencing any positive clinical response [60,61]. Therefore, it has been suggested to start azathioprine in conjunction with prednisone in patients with moderate to severe symptoms followed by starting steroid tapering once the desired symptom control is attained [62,63]. About 70–80% of patients receiving either azathioprine as a monotherapy or in combination with steroids had remission or marked improvement of symptoms, with higher response rates occurring when azathioprine was started earlier in the disease course [61,64,65,66,67]. In a prospective randomized study on 34 myasthenic patients comparing prednisolone + azathioprine to prednisolone + placebo, a significantly lower prednisolone dose was required to maintain remission at 2 and 3 years, with fewer relapse rates in the azathioprine compared to the placebo groups [68]. Another prospective randomized study suggested that azathioprine use may allow for rapid prednisone tapering [69]. The typical starting dose of azathioprine is 50 mg daily for one week and then an increase by 50 mg/day each week up to a dose of 2–3 mg/kg daily or divided into two doses. Once remission is achieved, it is ideal to continue therapy for at least two to three years before attempting to taper or withdraw azathioprine with close follow-up to monitor for signs of relapse [70]. Potential adverse effects include flu-like symptoms, which may be experienced in up to 10% of patients, myelosuppression, abdominal discomfort, nausea, anorexia, hepatotoxicity, and pancreatitis [71,72,73]. Ten to 20 percent of patients may develop an idiosyncratic drug reaction, manifested by fever, malaise, and a picture reminiscent of sepsis shortly after starting azathioprine, which will necessitate immediate and permanent discontinuation of that medication [56]. Leukopenia and hepatotoxicity are the main toxicities of azathioprine and are associated with elevated levels of 6-thiouric acid and 6-methyl mercaptopurine levels, respectively. As absent or low activity of TPMT predicts higher incidence of leukopenia, an assessment of enzymatic TPMT activity is recommended to screen for homozygous mutations (complete absence of enzymatic activity), in whom azathioprine should not be used and heterozygous mutations (reduced activity), in whom a lower dose of azathioprine should be started with closer monitoring during dose escalation [74]. On the other hand, increased activity of TPMT predisposes to hepatotoxicity through increased levels of 6-methyl mercaptopurine (hypermethylation state) [59] (Figure 1). As allopurinol inhibits xanthine oxidase, it increases the activity of azathioprine metabolites, which may result in severe leukopenia; therefore, the dose of azathioprine is to be reduced to 25–50% of the typical dose if patients on allopurinol [59]. Azathioprine-related pancreatitis is manifested as abdominal pain, elevated amylase levels >3 times, and positive imaging findings, and was significantly associated with smoking in a large cohort of patients with inflammatory bowel disease [73]. Biweekly monitoring of full blood count, chemistry panel, and liver function test is recommended upon starting or increasing the dose of azathioprine; after the patient is on a stable dose for 6 weeks, the frequency of monitoring can be reduced to monthly for 3 months, followed by every 3 months [59,74]. Megaloblastic red cells and lymphopenia are common in patients who take azathioprine and indicate therapeutic effect and medication compliance [59]. Azathioprine dose is to be reduced with lymphopenia <500 per mm^3^ or when WBC count falls below 4000 per mm^3^, and to be discontinued if WBC count is less than 1500 per mm^3^ or absolute neutrophil count falls below 1000 per mm^3^ [71]. We recommend slow tapering of azathioprine and discontinuation over 1–2 years; abrupt discontinuation of azathioprine resulted in a clinical relapse in more than 50% of patients in a previous study [75].

### 3.3. Tacrolimus

Tacrolimus is a calcineurin inhibitor that suppresses T-cell activity and proliferation by blocking the transcription of cytokine genes including IL-2, a similar mechanism of action as cyclosporine but with stronger immunosuppressive effects [76,77,78,79,80]. It has a rapid therapeutic onset with clinical improvement that can be observed as early as 10–28 days after therapy initiation. Uncontrolled studies have demonstrated marked clinical improvement or remission in approximately 70–87% of patients receiving tacrolimus within the first year of therapy [81,82,83,84]. Although two controlled studies failed to demonstrate a steroid-sparing effect of tacrolimus at 6 and 12 months, several uncontrolled studies have shown a significant reduction in oral steroid requirements with long-term use [85,86,87,88,89]. A prospective unblinded randomized study in de novo MG patients on prednisolone reported that the addition of tacrolimus significantly reduced the number of treatments with plasma exchange and daily oral steroid dose at one year [87]. The starting dose of tacrolimus is 3 mg daily or 0.1 mg/kg/day in two divided doses with a target trough concentration of 4.8–10 ng/mL [81,90,91]. Side effects include hyperglycemia, hypomagnesemia, hypertension, headache, tremors, diarrhea, nausea, and paresthesias [92].

### 3.4. Mycophenolate Mofetil

Mycophenolate mofetil suppresses T- and B-lymphocytes proliferation by inhibiting the enzyme inosine monophosphate dehydrogenase involved in the biosynthesis of de novo guanosine nucleotides [93]. It was initially introduced as a therapeutic option in MG after several case reports, case series, and pilot studies revealed promising beneficial effects in MG patients with varying degrees of severity in the early 2000s [94,95,96,97,98,99,100]. Clinical improvement may be seen 5 months after therapy initiation, but may take up to 10–12 months, although earlier improvements have been reported [97,101]. In an earlier study, approximately 50% of cases treated with mycophenolate monotherapy or in conjunction with steroids showed marked clinical improvement (i.e., minimal manifestation status) within the first year of therapy and up to 73–80% after two years [102]. In a prospective observational study of 31 patients with ocular MG, 93% of patients remained purely ocular over a mean period of 4.2 years when prednisone was switched to mycophenolate [103]. Other studies have suggested a steroid-sparing effect of long-term mycophenolate use with steroid dose reduction expected in up to 68–75% of cases and steroid discontinuation in 13–55% of patients with generalized MG [101,102]. On the other hand, two randomized controlled trials have failed to demonstrate benefits of mycophenolate in conjunction with prednisone over prednisone alone [104,105]. These negative results could, however, be attributed to the effect of prednisone, even at a lower dose in the placebo group, which delayed therapeutic onset of mycophenolate given the short duration of the studies and the milder disease status in the selected subjects [104,105,106,107]. The typical dosage of mycophenolate mofetil is 2–3 g/day in two divided doses. Once the therapeutic effect is attained, it is suggested to continue mycophenolate for a few years before attempting to slowly taper the dose by no more than 500 mg/day annually to prevent symptom relapse or MG exacerbations [108]. Some of the potential side effects include nausea, vomiting, diarrhea, leucopenia, and opportunistic infections [109,110]. Mycophenolate mofetil may be considered in patients who cannot tolerate steroids, where steroid therapy is contraindicated, or when steroid-sparing effects are desired. In our practice, we have been using mycophenolate less commonly as a steroid-sparing agent.

### 3.5. Cyclosporine

Cyclosporine is a calcineurin inhibitor with a similar mechanism of action to tacrolimus. Several studies have demonstrated the beneficial effects of cyclosporine in treating myasthenic patients and reducing steroid requirements [111,112,113]. However, due to its safety profile, it is not commonly used by the authors. Clinical improvement is expected within the first 2 months of therapy initiation with maximal improvement apparent after a median of 7 months [114]. The suggested dose of cyclosporine is 5–6 mg/kg/day in two divided doses. Possible side effects include flu-like symptoms, myalgia, nephrotoxicity, hypertension, gingival hyperplasia, hypertrichosis, postural tremor, headache, paresthesias, and optic neuropathy [115,116,117]. Due to its nephrotoxicity, it is imperative to monitor renal function with routine serum creatinine checks, particularly within the first few months of therapy.

### 3.6. Methotrexate

Methotrexate is a folic acid antagonist, its immunosuppressive effect is partly due to the inhibition of dihydrofolate reductase, and decreasing the activation of nuclear factor- κB, therefore preventing the conversion of dihydrobiopterin to tetrahydrobiopterin, which ultimately results in an increased T cell apoptosis [118]. Methotrexate, when administered at 20 mg oral every week, did not show a steroid-sparing effect over a period of 12 months in a multicenter randomized, placebo-controlled study [119]. However, significantly more patients in the placebo arm dropped out of the study due to worsening symptoms, and a post hoc analysis has shown better outcomes in MG activity of daily living (MG-ADL) and quantitative MG score (QMG) in the methotrexate arm scores [56,119]. Furthermore, another study demonstrated favorable efficacy of methotrexate as a steroid-sparing drug in a cohort of patients who had not benefitted or had side effects from azathioprine [120]. However, the results of the latter study are to be interpreted with caution due to the retrospective nature and the small sample size. Side effects of methotrexate include hepatotoxicity (elevation of liver enzymes in 10–43%), pulmonary fibrosis, cytopenia, renal insufficiency and dermatological side effects, especially oral ulcers [121].

### 3.7. Cyclophosphamide

Cyclophosphamide is non-phase-specific alkylating agent that acts on DNA and inhibits rapidly proliferating cells such as T- and B-lymphocytes [122,123]. Several uncontrolled studies have demonstrated the beneficial effects of cyclophosphamide in patients with severe, steroid-dependent, or refractory MG, with rapid clinical improvement observed in up to two-thirds of patients within the first month of therapy [123,124,125]. In one randomized controlled study of 23 myasthenic patients with steroid-dependent moderate to severe disease, those who were treated with cyclophosphamide had significant clinical improvement at one year and required a lower steroid dose at 6 and 12 months as compared to placebo [126]. Reported treatment protocols include either monthly IV pulse dosing of 500 mg/m^2^ or 1–2 mg/kg orally per day [124,126]. Ablation of the immune system by high-dose cyclophosphamide (50 mg/kg/day intravenously for 4 days) followed by treatment with granulocyte colony stimulating factor was shown to be a safe and effective treatment for MG patients who have failed multiple immunomodulatory treatments [18,127]. Eleven of 12 refractory MG patients treated with the high-dose cyclophosphamide had a dramatic response from 5 months to 7.5 years [18]. Side effects include hemorrhagic cystitis, alopecia, myelosuppression, infection, nausea, and vomiting [128].

### 3.8. Hematopoietic Stem Cell Transplantation (HSCT)

A 17-year-old male with AChR Ab+ MG refractory to multiple treatments including PLEX, IVIG, rituximab and cyclophosphamide pulses had a complete clinical remission except for persistent ophthalmoplegia, after allogenic HSCT from an HLA-matched sibling [129]. In another report, seven treatment refractory MG patients underwent ablation of the autoreactive immune cells using high-dose chemotherapy +/− total-body irradiation and antilymphocyte antibodies, followed by reconstitution of the immune cells with previously harvested autologous stem cells that were depleted of residual mature immune cells through CD34 immunomagnetic selection (autologous HSCT) [19]. All patients in that study had complete and stable remissions off maintenance immunomodulatory treatment, over a median follow up of 40 months. HSCT requires long hospitalization, a mortality rate of 6–8%, potentially severe short-term side effects such as mucositis, neutropenia, opportunistic infections, as well as late complications including regimen-specific organ toxicity and the emergence of malignancy and secondary autoimmune disease [130].

## 4. Biologicals: Monoclonal Antibodies Targeting Immune System

### 4.1. Rituximab

Rituximab is a humanized chimeric monoclonal antibody directed to CD-20, resulting in complement-mediated cytotoxicity and, therefore, depletion of CD-20+ cells, preventing B-cell activation and proliferation [131]. Earlier studies consisting mostly of case reports have demonstrated the beneficial effects of rituximab in patients with refractory or severe generalized MG [132,133,134,135,136,137]. Additionally, uncontrolled studies have shown that rituximab use allows for significant reduction in or cessation of steroid and other immunosuppressants [138,139]. The efficacy of rituximab is more evident in MG with antibodies to muscle-specific tyrosine kinase (MuSK Ab+) compared to those with antibodies to acetylcholine receptor (AChR Ab+), with 70–89% achieving minimal manifestation status (MMS) or better in MuSK Ab+ MG versus with 30–47% in AChR Ab+ MG and remission rate of 47% versus 16%, respectively [140,141,142]. In a prospective open label study on a cohort of MuSK Ab+ MG patients who were followed up for a median duration of 3.5 years, 58% of patients who received rituximab vs. 16% of those who did not had a favorable clinical outcome and lower doses of immunosuppressants [143]. Rituximab was also shown to be safe and effective in a cohort of treatment refractory late-onset AChR Ab+ patients [144]. On the other hand, in a placebo-controlled double blinded study on 52 AChR Ab+ MG patients, rituximab did not meet the primary end point of showing significant steroid-sparing effect, as assessed by the proportion of patients achieving a > 75% decrease in mean daily prednisone dose one year after starting the treatment [145]. However, the patients on the placebo arm had a three-fold higher relapse rate compared to the rituximab group, and the study could have been underpowered. Commonly reported treatment protocols include 375 mg/m^2^ weekly for 4 weeks or two infusions of 1000 mg doses given two weeks apart, although lower doses have also demonstrated effectiveness [140,146,147]. Repeated cycles may be administered at a 3–6-month interval if clinically indicated or repopulation of CD19+ CD27+ (rather than total CD19+) memory B cells [148,149]. Side effects include infusion reactions (pruritis, flushing, dyspnea, and chills), infection, hematological disorders, alopecia areata, and paroxysmal atrial fibrillation [149,150]. As rituximab has resulted in severe hepatitis B virus (HBV) reactivations not only in HBsAg-positive patients but also in HBsAg-negative and anti-HBc-positive patients [151], patients who are to receive rituximab should be screened for HBV infection by testing both HBsAg and HBcAb, and those who test positive should be treated with antiviral prophylaxis, usually lamivudine [152,153]. Rituximab has also been associated with reactivation of hepatitis C infection and tuberculosis, necessitating serological assessment before treatment initiation [153]. Age-appropriate inactivated vaccinations (i.e., influenza, pneumococcal) and vaccination for SARS-CoV-2 should be provided prior to starting the rituximab treatment; on the other hand, live-virus vaccines (i.e., VZV, measles–mumps–rubella) should be avoided for at least 6 months after completion of rituximab treatment [153].

### 4.2. Eculizumab

Eculizumab is a monoclonal antibody that targets the C5 protein in the complement cascade, inhibiting its cleavage and subsequently preventing the release of proinflammatory mediators and the formation of membrane attack complex (MAC), thus decreasing complement-mediated damage at the NMJ [154]. The safety and efficacy of Eculizumab for treatment refractory MG was assessed in a phase III, a randomized double-blind placebo-control study on patients with refractory MGFA class II-IV AChR Ab+ generalized MG (REGAIN trial). Patients in the study were randomized to either eculizumab or placebo for 26 weeks, which demonstrated significant improvement in MG-ADL and QMG scales in patients treated with eculizumab, with about 60% of patients achieving improvement and 25% achieving minimal manifestation vs. 41% and 13%, respectively in the placebo group [155,156]. Moreover, an extended open label study with a median follow-up of 22.7 months showed sustained efficacy of eculizumab, with ~50% of patients achieving minimal manifestations status or pharmacological remission [156,157]. Whether eculizumab is effective in the treatment of MG crisis is not established; a rapid and significant improvement in respiratory status after treatment with eculizumab was reported in three patients with refractory MGFA class V ventilator-dependent MG who have failed other conventional therapy including PLEX and IVIG, leading to extubation in two of the patients [158]. The reported regimen include 900 mg given weekly for 4 weeks and then 1200 mg at week 5, and every two weeks thereafter [159]. Common side effects include headache, nausea, diarrhea, upper respiratory tract infection, nasopharyngitis, and arthralgia [157]. It is recommended to administer the meningococcal (polyvalent + meningococcus B) vaccine at least two weeks prior to eculizumab initiation, followed by vaccine booster one month later, since complement inhibition may increase the risk of infection with encapsulated bacteria such as Neisseria meningitides [160].

### 4.3. Efgartigimod

The neonatal fragment crystallizable (Fc) receptor (FcRn) is expressed by a variety of cells and tissues including the endothelial cells, and plays a role in recycling of IgG, therefore increasing its half-life in the circulation through decreasing its degradation in the lysosomes [161]. FcRn blockers accelerate the degradation, thus decreasing the circulating levels of pathogenic IgGs and therefore downstream pathogenic events such as complement activation. Although FcRn blockers decrease the level of non-pathogenic IgG as well, they have a limited risk of predisposition to serious infections as they do not decrease the levels of IgA or IgM, nor impair the production or the quality of IgGs [162]. A number of FcRn blockers are therefore in different stages of development for MG and other antibody-mediated diseases [162], with Efgartigimod receiving FDA approval for AChR Ab+ MG in 2021. Efgartigimod consists of the Fc portion of IgG1, engineered to increase its affinity to the IgG binding site of FcRn [163]. In a phase 2 study, efgartigimod was administered at 10 mg/Kg intravenously every week as four doses to 12 AChR Ab+ patients [164]. Efgartigimod resulted in reduced levels of AChR Abs to 40–70% of the baseline starting at day 15 after the first dose and sustained reduction was present to day 29, in all but one patient [164]. Maximal improvement in MG-ADL and quantitative MG scores occurred 1–2 weeks after the 4th infusion. A follow up phase 3, randomized, placebo-controlled (ADAPT) study showed improvement of >2 points in MG-ADL after the first infusion cycle in 68% of those who received efgartigimod vs. 30% of those who received placebo [159]. About one-third of the patients who responded to efgartigimod needed a second set of four weekly infusions after 6–7 weeks, whereas the therapeutic efficacy, as defined by decrease in MG-ADL of >2 compared to the baseline, persisted for 12 weeks or more in another third. Although patients with MuSK Ab+ and double negative MG were included in the ADAPT study, the study was not powered to prove the effectiveness for those MG subtypes. Efgartigimod was overall very well-tolerated with the most common side effect being headaches, but its prevalence was like the placebo. Infections, which were mostly mild to moderate in severity, were noted in 46% of patients in the efgartigimod arm compared to 37% of those on the placebo.

## 5. Treatments Used in Severe MG Exacerbation or Crisis

### 5.1. Intravenous Immunoglobulin (IVIg)

IVIg was introduced as a therapeutic modality for MG nearly four decades ago, following several reports that demonstrated positive response in the management of MG patients with severe disease or acute exacerbation [12,13,14,165]. Two randomized studies have suggested that IVIg is equal or comparable in efficacy to plasma exchange, but with a better safety profile during acute exacerbations [166,167]. Moreover, noncontrolled studies have demonstrated the beneficial effects of long-term IVIg use in reducing symptom burden, maintaining remission or symptom control, and providing a steroid-sparing effect in patients with generalized MG [168,169,170]. IVIg has a rapid therapeutic onset with symptom improvement observed within days, maximal response 7–10 days after therapy initiation, and treatment effects lasting 28–60 days [171]. Positive clinical response is expected in 70–90% of patients receiving IVIg within the first two weeks of therapy, with higher response rates and longer duration of improvements in those who are on other immunosuppressive therapies or have had a thymectomy [165,172,173]. The typical IVIg dose for an acute MG exacerbation is 2 g/kg divided over a period of 3–5 days, with the most commonly used maintenance dose 0.4 g/kg given as a single dose every 3–6 weeks [169,174]. Side effects include headache, rash, myalgia, chills, fever, shortness of breath, and nausea [175,176]. Other uncommon side effects are aseptic meningitis, acute renal failure, and thromboembolic events [177,178,179]. Caution is to be exercised in the administration of IVIG to older patients with abnormal renal function at baseline; small doses and sucrose-free IVIG brands are preferred in those patients with close monitoring of the kidney function [179,180]. Although IVIg is generally used for patients with MG exacerbation or crisis, maintenance IVIg may be considered in patients who have failed to attain optimal symptom control while on conventional immunosuppressants [168,169,174]. IVIG infusion is also commonly used to stabilize patients before surgery and as bridging therapy during the initiation of high-dose steroids to minimize or prevent the paradoxical worsening of underlying weakness, which is particularly of concern in those with bulbar or respiratory involvement [55,181].

### 5.2. Plasma Exchange (PLEX)

PLEX was first introduced for MG in 1976, when it was effective for two patients with MG refractory to AChE inhibitors, steroids and thymectomy [16]. The efficacy of PLEX for MG is through direct removal of pathogenic autoantibodies and compliment pathway components and changes in the cytokine profile such as increased level of interleukin 10 [182,183,184]. PLEX is now frequently used as one of the first line acute treatment modalities (the other being IVIG) in MG crisis or in preparation for surgical interventions such as thymectomy in MG patients with bulbar and respiratory symptoms [181,182]. PLEX is preferred by some over IVIG in critically ill patients in MG crisis because of its faster therapeutic effect, which may be noted as early as 3 days after starting of its administration [185,186]. Several studies have shown comparable efficacy of PLEX and IVIG when assessed beyond the first week of administration. PLEX, when administered at 1 plasma volume for 5 sessions was equally effective and tolerated as IVIG 2 g/kg within 2 weeks of treatment in a randomized study on patients with moderate to severe MG with a mean QMG score >10.5 [167]. A prospective, open label study on 10 AChR Ab+ MG patients who received 5–6 sessions of PLEX (1 plasma volume/session) demonstrated significant improvement of MG-ADL, MG-manual motor test, and quality of life-15 (MG-QoL15) after 2 weeks, with maximal improvement in 6–12 weeks [187]. Only 2 of 58 patients who received PLEX did not have a significant response in another retrospective study involving AChR Ab+, MuSK Ab+ and double seronegative MG, with male sex and late onset MG being the predictors of a better response [188]. On the other hand, in a cohort of juvenile MG patients who received PLEX, IVIG, or both, the PLEX had significantly more consistent efficacy, as all of the 7 patients who underwent PLEX and only 50% of 10 who had IVIG treatment responded favorably [189]. The findings of that study are to be interpreted with caution because of the small number of patients. PLEX is also used in a periodic manner in patients refractory to multiple treatment modalities including immunosuppressants and IVIG [190]. The choice between PLEX vs. IVIG in the acute treatment of MG exacerbation and crisis partly depends on the expertise of the treating facility in the administration of PLEX and insurance coverage of IVIG [56]. PLEX can be administered as outpatient and with peripheral access, and in that setting, the adverse effects are rather mild [191]. On the other hand, PLEX is often considered a complex treatment when there is a need for central venous access and hospitalization. Inpatient PLEX usually necessitates insertion of a central venous catheter, which is inserted in a vein in the upper chest or neck, and terminates at the junction of superior vena cava and right atrium [192]. Patients who are to undergo successive, maintenance PLEX usually will need a tunneled central venous catheter, an arteriovenous fistula, or a double lumen port [56,192]; the latter is favored by the authors because of the lower infection rate with long-term use. Complications of PLEX include those due to central catheters and those related to the procedure. Central line placement complications include pneumothorax, line infection and thromboembolism [56]. The authors have seen a patient with bilateral pneumothorax which occurred after a central line placement in a patient with history of thymectomy [193]. Other possible complications of PLEX include citrate reaction due to hypocalcemia, which can cause perioral and limb paresthesia, nausea, vomiting, and rarely tetany, seizures and hypotension, fever, coagulopathy, and allergic reactions [194]. Citrate toxicity can be prevented by adding of 0.5 g of 10% calcium gluconate to the albumin-containing replacement fluid [191].

## 6. Thymectomy

The role of the thymus and thymic malignancies in the pathogenesis of MG has been discussed in another manuscript in this Special Issue [1]. Thymectomy was one of the first recognized therapeutic approaches for MG and is mandatory if a thymoma is present [5,195]. The efficacy of thymectomy for non-thymoma-related MG was demonstrated in a multicenter, international, randomized, rater-blinded study (MGTX study) in patients with AChR Ab+ MG of 18–65 years of age, and MG duration of less than 5 years [7]. MGTX study compared thymectomy vs. no thymectomy + alternate day prednisone in both arms, over a period of 3 years. Patients who received a thymectomy had a significantly better QMG score, less hospitalizations due to MG crisis, and required lower prednisone dose or adding azathioprine to achieve and maintain MMS. The superiority of the thymectomy arm in regard to lower QMG score and prednisone dose was apparent in 3 months and persisted for the 3 years. Follow-up of about 60% of the subjects in the MGTX study demonstrated the continued efficacy of thymectomy for another 2 (total period of 5) years [196]. All of the patients in the MGTX study received an extended, trans-sternal thymectomy, which removes 85–95% of the thymic tissue [7]. The procedure is, however, associated with longer intubation time and postoperative hospital stay, which, along with cosmetic reasons, results in hesitance on the part of MG experts and patients to proceed with the procedure [197]. Complication rate is generally higher in older patients who undergo extensive surgical procedures and only 8 of 66 patients who received thymectomy in the MGTX study were older than 50; the postoperative complications were not discussed in that study. Less invasive procedures, including transcervical thymectomy, video-assisted thoracoscopic surgery (VATS), and robotic VATS are associated with a faster recovery, shorter hospital stay, and better cosmetic outcomes. Minimally invasive procedures, especially robotic VATS, are therefore being increasingly used worldwide in the last decade [197,198]. Another study suggested that thymectomy was associated with complete remission in about 60% of MG patients with thymic hyperplasia, with better outcomes in patients below the age of 40 and disease duration of less than 12 months [199]. As with any major surgical procedure, thymectomy may elicit MG exacerbation or crisis and, therefore, should optimally be performed in a stable MG status. Patients with low respiratory reserves or bulbar symptomatology are to be treated with IVIG or PLEX before the thymectomy procedure [56]. The effectiveness of thymectomy for non-AChR Ab+ MG variants is not supported by the current evidence [181]. In a post hoc analysis of patients with MuSK Ab+ MG who were part of a study assessing the efficacy of rituximab, there was no significant difference in patients who had thymectomy vs. those who did not, regarding achievement of an MMS [200].

Table 1 summarizes the class of evidence, overall efficacy, common or important adverse effects, and level of recommendation for the different immunomodulatory treatment options of MG discussed above.

## 7. Treatment Strategy of MG

MG is a chronic autoimmune disease, although spontaneous remissions can occur, i.e., a quarter of patients before the use of immunosuppressants had spontaneous remissions for 4–17 years [4,214]. The treatment strategy of MG is based on disease severity, i.e., ocular vs. generalized, and if the latter, the severity of symptoms, especially whether the patient is in exacerbation or crisis. Another factor to be considered is the MG subtype from the serological perspective (see Figure 2). The goal of treatment is attaining a complete remission (no symptoms or signs of MG) or an MMS as defined by no symptoms but mild weakness in some muscles on exam, mostly noted in orbicularis oculi or hip flexors [56,181]. However, a large proportion of the MG patients fail to attain a complete and stable remission and about 10–13% of MG patients are refractory or intolerant to different treatment modalities [215,216,217]. Female and MuSK Ab+ patients are more likely to have a refractory phenotype [216,218]. Improving the function and quality of life of treatment refractory patients remains to be a more realistic goal than achieving an MMS. The ideal immunomodulatory treatment should have a favorable adverse effect profile, either no symptoms or mild symptoms, which do not need any intervention, i.e., a grade 1 or lower in the Common Terminology Criteria for Adverse Events (CTCAE) [181].

Some of the ocular MG patients, especially those with intermittent ptosis and mild diplopia, are treated with oral pyridostigmine only. The authors sometimes use terbutaline 2.5 mg 3 times a day in selected cases who do not tolerate or lose the response to pyridostigmine for both ocular and generalized cases (see Section 2.2). Ocular MG patients who fail on pyridostigmine generally respond very well to oral prednisone. In a previous prospective study, only 17% of ocular MG patients did not respond to up to 40 mg/day of oral prednisone and the median time to minimal manifestation status was 14 weeks [48]. The dose of prednisone should be slowly tapered after achieving remission or minimal manifestation status (see Section 3.1), and if there is relapse of the symptoms with doses above 7.5–10 mg/day, or 15–20 mg every other day, use of a steroid-sparing drug is indicated [35]. Ocular MG is rarely refractory to the treatment with steroids and other oral steroid-sparing drugs; some of the refractory cases have been successfully treated with IV methylprednisolone, IVIG, as well as nonpharmacologic treatments such as occlusive devices, prisms, eyelid supports, eyelid lift and strabismus surgery [219,220].

Patients with mild generalized MG symptoms may only be treated with pyridostigmine, but most would need immunomodulatory treatment. It should be noted that pyridostigmine is often not effective or is poorly tolerated in patients with MuSK Ab+ MG [181,221]. We recommend thymectomy to patients with thymoma (mandatory), AChR Ab+ patients with generalized MG who are <50 years old and disease onset <5 years. Thymectomy is not considered for MuSK Ab + patients and its efficacy is not established for double seronegative patients. Prednisone is usually the first line of immunosuppressant treatment in generalized MG, with the treatment schedule depending on the clinical scenario. Given the possibility for paradoxical worsening, patients with poor bulbar and respiratory reserve should be treated first with PLEX or IVIG before starting a high dose of prednisone (40–60 mg/day). We do not have experience with high-dose IV methylprednisolone for the treatment of MG, but it has been reported in case studies or small cohorts of generalized and even ocular MG patients [207,219,222,223,224]. After attaining a remission or MMS, a slow taper is started (see Section 3.1). If a high dose of prednisone is needed to prevent a relapse, a steroid-sparing agent is usually added. The authors generally use tacrolimus 2–4 mg/day, once per day for a long-acting formulation, otherwise divided to two doses per day, as the first line steroid-sparing agent. We advise the patient to monitor BP and periodically check kidney function, and do not monitor tacrolimus trough levels. Alternatives to tacrolimus include azathioprine, and less often mycophenolate, or cyclosporine; we generally do not use methotrexate, and have only rarely used cyclophosphamide. We have used rituximab in AChR Ab+ in patients who had a concomitant lymphoproliferative disease, which resulted in complete remission of MG [204]. Non-steroid immunosuppressants can be used as the first-line monotherapy in patients who are poor candidates of steroid treatment, such as patients with severe diabetes, peripheral edema, or obesity, but may be require IVIg while waiting for the therapeutic effect to occur. We do not recommend using more than one oral immunosuppressant plus steroids because of increased risk of immunosuppression-related side effects; one of the authors has seen a patient with CMV colitis when being treated with prednisone, azathioprine, and mycophenolate. Patients who lack good symptom control on two oral immunosuppressants or are on one immunosuppressant but need frequent use of IVIG or PLEX (more than four per year) are considered treatment refractory [155]. We have used eculizumab (only in AChR Ab+ cases), rituximab (mainly in MuSK Ab + patients), maintenance IVIG (0.5–1 g/kg every 2–4 weeks) and less frequently, maintenance PLEX (one session every 1 to 4 weeks) in treatment refractory cases. Efgartigimod is likely effective in different types of MG but the ADAPT study was powered to show efficacy only in AChR Ab + patients and is currently approved for that subtype [206]. Efgartigimod may be used in MG patients with significant symptoms (patients with MG-ADL score > 5 were included in the ADAPT study) regardless of status of steroid and immunosuppressive treatment [206]. Although eculizumab and efgartigimod are FDA-approved for generalized AChR Ab+ MG, their use is not currently widespread, largely because of their cost. Due to potentially serious adverse effects, cyclophosphamide and HSCT should only be considered in patients with refractory, life-threatening MG; the use of these will likely become more limited with the increased availability of targeted immunosuppressive therapy.

Figure 2 summarizes the overall treatment approach of MG, based on the clinical phenotype and serology.

## 8. Conclusions

Tremendous progress has been made in the treatment of myasthenia gravis in the last eight decades, making it one of the most treatable autoimmune diseases in humans. Although a minority of myasthenic patients have a spontaneous remission or respond to acetylcholinesterase inhibitors, most need treatment with steroids and/or steroid-sparing drugs. A small but significant proportion of MG patients remain refractory, lack tolerance, or develop side effects to steroids and immunosuppressants. Therefore, there is an unmet need for targeted immunomodulatory drugs, which has resulted in an ongoing campaign to develop safer and more effective treatments for myasthenia gravis. The recent development of biologicals, which have a more targeted mechanism of action and more favorable side effect profiles, may change the treatment algorithm of MG treatment in the future.

## Figures and Tables

**Figure 1 jcm-11-01597-f001:**
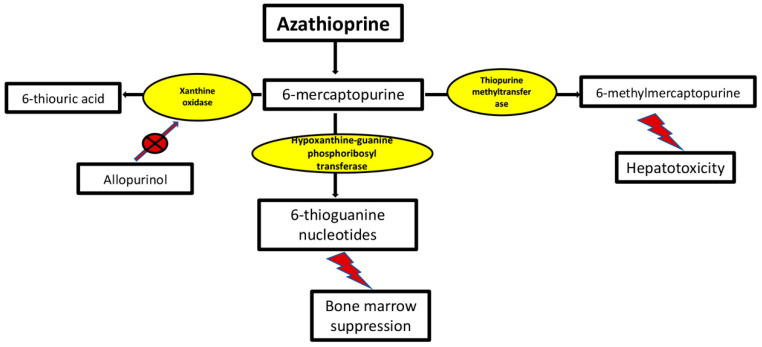
Azathioprine metabolism. Deficiency of thiopurine methyltransferase results in increased 6-thioguanine levels, which may result in myelosuppression. On the other hand, increased activity of thiopurine methyltransferase will lead to increased levels of 6-methylmercaptopurine, which predisposes to hepatotoxicity (hypermethylation). Allopurinol can result in severe leukopenia if administered with the usual dose of azathioprine as it inhibits xanthine oxidase.

**Figure 2 jcm-11-01597-f002:**
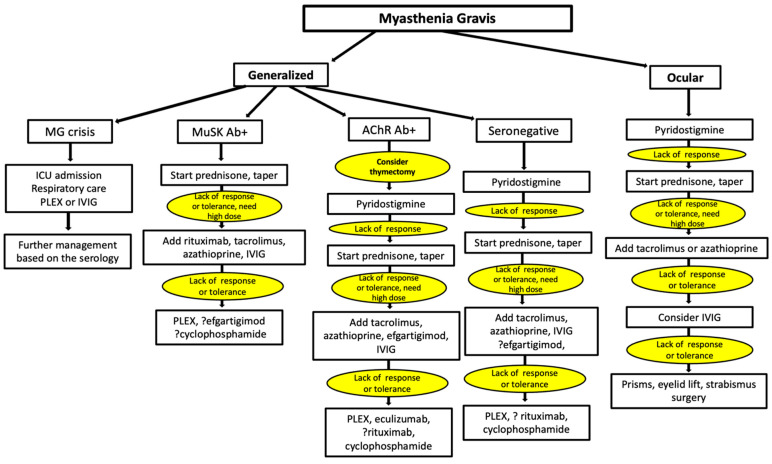
Overall treatment strategy of myasthenia gravis. MuSK: muscle specific tyrosine kinase; Ab: antibody; AChR: acetylcholine receptor; IVIG: intravenous immunoglobulin; PLEX: Plasma exchange.

**Table 1 jcm-11-01597-t001:** The class of evidence, overall efficacy, common and more significant side effects of immunomodulatory treatment options in MG.

	Class of Evidence (Supportive Studies)	Overall Outcome	Adverse Effects	Level of Recommendations
Prednisone	II [48,49,50,51,54,65,68,201,202]	Generally effective in ocular and generalized MG	Weight gain, edema, hypertension, hyperglycemia, osteoporosis, cataracts, infections, neuropsychiatric symptoms	Ocular and generalized MG who do not respond to pyridostigmine (level B). Monotherapy in selected patients if they are controlled by a low dose (level B)
Azathioprine	II [60,61,62,63,64,65,67,68,69,70]	Effective as a steroid-sparing agent	Leukopenia, hepatotoxicity, pancreatitis, sepsis like idiosyncratic reaction	MG not controlled with low steroid dose (level B)
Tacrolimus	II [81,82,83,84,85,86,87,88,90,92]	Effective as a steroid-sparing agent	Well tolerated in doses used for MG. Hypertension, nephrotoxicity, hyperglycemia, hypomagnesemia, tremors, diarrhea, nausea	MG not controlled with low steroid dose (level B)
Mycophenolate mofetil	II [94,95,97,98,99,100,101,102,103,104,105,203]	Although earlier results were promising, a subsequent large RCT did not prove steroid-sparing effects, which was attributed by some to issues with the study design, such as inadequate length of the study	Leukopenia, diarrhea, nausea, vomiting, hyperglycemia, headaches	MG not controlled with low steroid dose (level C)
Cyclosporine	II [111,112,113,114]	RCT supports the use of cyclosporine, but toxicity more frequent than for tacrolimus.	Nephrotoxicity, hepatotoxicity, hypertension, hypertrichosis, gingival hyperplasia, tremor, optic neuropathy	Level B recommendation, but use is limited by toxicity
Methotrexate	II [56,119,120]	Although a large RCT did not prove a steroid-sparing effect, a post hoc analysis suggested some efficacy in secondary endpoints	Hepatotoxicity, pulmonary fibrosis, infection	Insufficient evidence to recommend use (level U)
Cyclophosphamide	II [18,124,125,126]	Effective in patients with refractory generalized MG, including steroid-sparing effects	Bone marrow suppression, hemorrhagic cystitis, alopecia, infections, infertility, nausea and vomiting, neoplasia	MG refractory to other treatments (Level C), concern regarding severe adverse effects, studies conducted before the introduction of newer targeted therapies
Rituximab	II [132,133,134,135,136,137,138,139,140,141,142,143,144,145,146,147,149,150,204]	Efficacy more pronounced in MuSK Ab+, but also has shown efficacy and steroid-sparing effects in treatment refractory AChR Ab+ MG. A double blind RCT of rituximab did not prove steroid-sparing effect in AChR Ab+ MG but some have attributed the negative results to the design of the study	Well-tolerated in MG cases. Infusion-related reactions, hypotension, infections, leukopenia, thrombocytopenia, alopecia areata	MuSK Ab+ MG (level B), treatment refractory AChR Ab+ MG (level C)
Eculizumab	I [155,156,157,205]	Effective in refractory AChR Ab+ generalized MG, with long term steroid-sparing effects	Well-tolerated. Infusion-related reactions, severe meningococcal infection, other infections, headaches, musculoskeletal pain	Treatment refractory, highly symptomatic AChR Ab+ MG (level B), widespread use limited because of the price.
Efgartigimod	I [164,206]	Effective in generalized MG patients who remain highly symptomatic after treatment with pyridostigmine, steroids or NSI	Well-tolerated. Allergic reactions, headache, infections, leukopenia, myalgia	Level B recommendation for patients still symptomatic on pyridostigmine, steroids or NSI. Only approved for AChR Ab + MG, but may work for other MG subtypes, widespread use may be limited because of the price
IVIG	II [12,13,14,55,165,166,167,168,169,172,173,174,175,207,208,209,210]	Effective in MG exacerbation and crisis, and in refractory generalized MG, including long term steroid-sparing effects	Headache, urticaria, nephrotoxicity, thrombotic events, myalgia, fever, flu like symptoms	MG exacerbation or crisis (level B); maintenance therapy in refractory generalized MG (level C); in association with starting steroids or NSI (level C); widespread use limited because of the price
PLEX	II [16,166,167,185,187,188,190,209]	Effective in MG exacerbation and crisis, and in refractory generalized MG	Line infection, pneumothorax, hypocalcemia, hypotension, fever, coagulopathy, allergic reactions	MG exacerbation or crisis, (level B), maintenance therapy in refractory generalized MG (level C); use could be limited by availability of expertise and sometimes by need for central venous access
Thymectomy	II [7,196,199,200]	Effective in AChR Ab+ patients 18–65 years of age, including steroid-sparing efficacy. Not effective in MuSK Ab+ MG	Surgical complications, postoperative MG exacerbation	Must be carried out in MG with thymoma (level A); Recommended for 18–50-year-old, non-thymomatous AChR Ab + (level B), Not recommended in MuSK Ab + MG; inadequate evidence in double seronegative MG (level U)

Class of evidence is based on guidelines proposed by “2017 AAN Clinical Practice Guideline Process Manual” [211]. Levels of recommendation: A, effective, must be offered; B, probably effective, should be offered; C, possibly effective, may be offered; U, evidence is insufficient to support or refute the use [211,212,213]. NSI, non-steroid immunosuppressant; RCT, randomized clinical trial.

## Data Availability

Not applicable.

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
