# Peer review of "Current Treatment of Myasthenia Gravis"

_jcm, 2022, doi:10.3390/jcm11061597_

Round 1

Reviewer 1 Report

This is a very detailed and well-written review on the treatment of MG. They covered all the important points.

I have a few comments:

Twenty to 75% of MG patients who are started on steroids may experience varying degrees of transient worsening of weakness that may develop within the first two weeks of therapy initiation [53-55], …” Those which report a high rate of steroid-induced exacerbation are older studies with only a few patients (Brunner and Jenkins each reported 9 patients). I wonder if such high rates as 75% reflect the actual situation.

“Side effects are due to stimulation of ACh muscarinic and nicotinic receptors, which include gastrointestinal disturbances (abdominal cramps, diarrhea, nausea, increased salivation), increased bronchial secretions, lacrimation, hyperhidrosis, bradycardia, muscle cramps, and twitches.” It would be better to list the muscarinic effects saparately and clearly since they respond to treatment with anticholinergics. It is confusing in this sentence.

“…initial prednisone dose of 5-10 mg daily with a gradual dose escalation of 5 mg every 7-10 days up to 60-80 mg daily or until desirable symptom control is reached." Unless they are hoping for improvement at a lower dose, it would take about 6 weeks to get to 40 mg and many weeks to get to 60 mg on this regimen. Is this the message the authors want to give?

Mycophenolate mofetil: One of the main reasons offered for the negative results was the strong effect of even low dose steroids.

Azathioprine: It would be informative to give an idea about how it should be reduced, as has been done for mycophenolate mofetil (500 mg/day annually).

“…up to 80% of the MG patients fail to attain a complete and stable remission.”  This seems to be high. Much lower rates have ben given in several studies.

A few errors which I noted:

“Treatment of MG remains to be challenging as a subgroup of patients who are treatment refractory, therefore having recurrent hospitalizations for MG crisis, requiring maintenance IVIG or PLEX.” This sentence does not seem to be correct grammatically. Perhaps deleting ‘who’ could improve it.

Is ref 26 in the correct location?

Line 155: of

Author Response

Please find attached word document.

Sincerely,

Mohammed 

Reviewer 2 Report

The manuscript illustrates therapies of MG from non-immunosuppressive treatments to immunosuppressants, monoclonal antibodies, IVIg, PLEX, and thymectomy. These treatment summaries with historical background and practical doses are helpful. The authors also add information on adverse effects. The original figures are informative and illustrate their therapeutic concepts.

Strengths of the manuscript

  1. Covering broad therapeutic choices
  2. The manuscript includes the authors’ experiences and suggestions.
  3. The manuscript contains information on the adverse effects of therapies.

Weakness of the manuscript

  1. There is no information about the evidence level of each therapy.
  2. There is little information about the cost-effectiveness of therapies.

Major recommendations

1. Please provide a summary table of therapies, including information about types of clinical trials, endpoint, adverse effect, and evidence level

2. Reference 7 is inappropriate. Please replace with the reference below.

Wolfe GI, Kaminski HJ, Aban IB, Minisman G, Kuo HC, Marx A, Ströbel P, Mazia C, Oger J, Cea JG, Heckmann JM, Evoli A, Nix W, Ciafaloni E, Antonini G, Witoonpanich R, King JO, Beydoun SR, Chalk CH, Barboi AC, Amato AA, Shaibani AI, Katirji B, Lecky BR, Buckley C, Vincent A, Dias-Tosta E, Yoshikawa H, Waddington-Cruz M, Pulley MT, Rivner MH, Kostera-Pruszczyk A, Pascuzzi RM, Jackson CE, Garcia Ramos GS, Verschuuren JJ, Massey JM, Kissel JT, Werneck LC, Benatar M, Barohn RJ, Tandan R, Mozaffar T, Conwit R, Odenkirchen J, Sonett JR, Jaretzki A 3rd, Newsom-Davis J, Cutter GR; MGTX Study Group.N Engl J Med. 2016 Aug 11;375(6):511-22. doi: 10.1056/NEJMoa1602489.

Author Response

Please find the attached word document.

Sincerely, 

Mohammed 
